

# Mode of action of the 2-phenylquinoline efflux inhibitor PQQ4R against *Escherichia coli*

Diana Machado[1], Laura Fernandes[1,3], Sofia S. Costa[1], Rolando Cannalire[2], Giuseppe Manfroni[2], Oriana Tabarrini[2], Isabel Couto[1], Stefano Sabatini[2] and Miguel Viveiros[1]

[1] Unidade de Microbiologia Médica, Global Health and Tropical Medicine, GHTM, Instituto de Higiene e Medicina Tropical, IHMT, Universidade NOVA de Lisboa, UNL, Lisboa, Portugal
[2] Department of Pharmaceutical Sciences, Universitá degli Studi di Perugia, Perugia, Italy
[3] Current affiliation: Laboratório de Diagnóstico Molecular Veterinário GeneVet, Algés, Portugal

## ABSTRACT

Efflux pump inhibitors are of great interest since their use as adjuvants of bacterial chemotherapy can increase the intracellular concentrations of the antibiotics and assist in the battle against the rising of antibiotic-resistant bacteria. In this work, we have described the mode of action of the 2-phenylquinoline efflux inhibitor (4-(2-(piperazin-1-yl)ethoxy)-2-(4-propoxyphenyl) quinolone – PQQ4R), against *Escherichia coli,* by studding its efflux inhibitory ability, its synergistic activity in combination with antibiotics, and compared its effects with the inhibitors phenyl-arginine-β-naphthylamide (PAβN) and chlorpromazine (CPZ). The results showed that PQQ4R acts synergistically, in a concentration dependent manner, with antibiotics known to be subject to efflux in *E. coli* reducing their MIC in correlation with the inhibition of their efflux. Real-time fluorometry assays demonstrated that PQQ4R at sub-inhibitory concentrations promote the intracellular accumulation of ethidium bromide inhibiting its efflux similarly to PAβN or CPZ, well-known and described efflux pump inhibitors for Gram-negative bacteria and whose clinical usage is limited by their levels of toxicity at clinical and bacteriological effective concentrations. The time-kill studies showed that PQQ4R, at bactericidal concentrations, has a rapid antimicrobial activity associated with a fast decrease of the intracellular ATP levels. The results also indicated that the mode of action of PQQ4R involves the destabilization of the *E. coli* inner membrane potential and ATP production impairment, ultimately leading to efflux pump inhibition by interference with the energy required by the efflux systems. At bactericidal concentrations, membrane permeabilization increases and finally ATP is totally depleted leading to cell death. Since drug resistance mediated by the activity of efflux pumps depends largely on the proton motive force (PMF), dissipaters of PMF such as PQQ4R, can be regarded as future adjuvants of conventional therapy against *E. coli* and other Gram-negative bacteria, especially their multidrug resistant forms. Their major limitation is the high toxicity for human cells at the concentrations needed to be effective against bacteria. Their future molecular optimization to improve the efflux inhibitory properties and reduce relative toxicity will optimize their potential for clinical usage against multi-drug resistant bacterial infections due to efflux.

Corresponding author
Miguel Viveiros,
mviveiros@ihmt.unl.pt

# INTRODUCTION

The emergence of drug resistant bacteria represent a global threat to human health and is now a top priority of the World Health Organization (WHO) and the European Centre for Disease Prevention and Control (ECDC) programs for the treatment of infectious diseases. Particularly, the emergence of drug resistance in *Escherichia coli* requires close attention since the rate of isolates resistant to the commonly used antibiotics is rising worldwide (*World Health Organization, 2014*; *European Centre for Disease Prevention and Control, 2015*). The current therapeutic options are scarce to deal with these infections. Therefore, studies on new drugs and drug combinations as well as an improved understanding of the mechanism of action of these new drugs have become critical to fight the spread of multidrug resistant organisms.

In Gram-negative bacteria, besides the acquired resistance by the acquisition of external resistance determinants or mutations in genes that code for the drug targets, the intrinsic drug resistance also play an important role in the resistance towards antibiotics and biocides (*Viveiros et al., 2007*; *Piddock, 2006*; *Piddock, 2007*; *Nikaido & Pagès, 2012*). This resistance occurs as a consequence of the (i) presence of an outer membrane that create a permeability barrier reducing the influx of antimicrobials, and (ii) overexpression of efflux pumps that help to reduce the intracellular level of antimicrobials and toxins (*Nikaido & Pagès, 2012*; *Piddock, 2006*).

The efflux pumps of the RND (resistant nodulation cell division) superfamily have been clearly associated with multidrug resistant phenotypes in Gram-negative pathogens (*Nikaido & Pagès, 2012*). The substrates of the RND efflux pumps are different in their structure and physicochemical properties and include antibiotics, detergents, and biocides (*Piddock, 2006*; *Li & Nikaido, 2009*). The clinical implication of this substrate promiscuity is the development of multidrug resistance. The major RND efflux system of *E. coli* consists in a typical tripartite efflux pump, the AcrAB-TolC. This structure is composed by an integral membrane efflux transporter (AcrB), an outer membrane channel (TolC), and a periplasmic adapter protein (AcrA) (*Du et al., 2014*). Upon entering in the cell, the compounds will interact with the substrate-binding pocket of AcrB, which will extrude the compounds via TolC using the energy produced by the proton motive force (PMF) (*Nikaido & Takatsuka, 2009*). The AcrAB activity and overexpression have been associated with the resistance to fluoroquinolones, chloramphenicol, tetracycline, β-lactams, and β-lactamase inhibitors, among others, as well as biofilm formation and pathogenicity (*Piddock, 2007*). Beside the AcrAB-TolC efflux pump, other *E. coli* efflux systems also play a role in the development of drug resistance (*Viveiros et al., 2007*; *Nishino et al., 2003*). The overexpression of these efflux systems in response to the antibiotic stress is commonly the first step in the evolution of antibiotic resistance in the bacterial population, favoring the spontaneous appearance and stabilization of chromosomal mutations in the genes related

with the antibiotic action. This biological phenomenon leads to the emergence of resistance to almost all classes of antibiotics available and require the urgent implementation of novel therapeutic strategies for the management of these infections.

The inhibition of the efflux activity may be of great potential once used in combination therapies by restoring or enhancing the activity of the currently used antimicrobials and preventing the emergence of drug resistance. Several compounds capable of inhibiting *E. coli* efflux pumps have been described (*Lomovskaya et al., 2001*; *Viveiros et al., 2005*; *Bohnert, Schuster & Kern, 2013*; *Opperman et al., 2014*; *Vargiu et al., 2014*; *Yilmaz et al., 2015*; *Bohnert et al., 2016*; *Zuo, Weng & Wang, 2016*) but, to date, none has reached the clinical development mainly due to their toxicity. Moreover, the putative mechanisms of action of the great majority of the efflux inhibitors currently in study remain largely unknown. The most known *E. coli* efflux inhibitors are the peptidomimetic phenyl-arginine- β-naphthylamide (PAβN) and the phenothiazine chlorpromazine (CPZ). PAβN is a potent inhibitor of *Pseudomonas aeruginosa* efflux pumps (*Lomovskaya et al., 2001*) and is a substrate competitor of the AcrAB efflux pump of *E. coli* (*Viveiros et al., 2008*). Recently, *Misra et al. (2015)* showed that PAβN acts as an inhibitor of the AcrAB and AcrEF efflux systems at low concentrations, and as a membrane-destabilizing agent when used at higher concentrations. The phenothiazine CPZ presents efflux inhibitory activity against *E. coli* efflux systems (*Viveiros et al., 2005*). The phenothiazines mode of action involve the disruption of the calcium-calmodulin transport and signaling pathways (*Salih et al., 1991*; *Pluta, Morak-Młodawska & Jeleń, 2011*), modifications at the level of the bacterial membrane and on nucleic acid stability (*Pluta, Morak-Młodawska & Jeleń, 2011*; *Thorsing et al., 2013*), and inhibition of the type II NADH-ubiquinone dehydrogenase (*Weinstein et al., 2005*; *Schurig-Briccio et al., 2014*).

In recent studies, the 2-phenylquinoline derivative PQQ4R showed to be active as an efflux pump inhibitor of the Gram-positive *Staphylococcus aureus* (*Sabatini et al., 2011*; *Sabatini et al., 2013*) and non-tuberculous mycobacteria (*Machado et al., 2015*) but nothing is known about its mode of action. In this work, we aimed to unravel the mode of action of PQQ4R as efflux pump inhibitor, as in the search for new and effective efflux inhibitors is important to understand their inhibitory mechanism of action and to disclosure the presence of non-efflux related mechanisms (*Venter et al., 2015*). Because drug efflux is energy dependent, we hypothesize that the mode by which PQQ4R inhibits drug efflux involves the interference with the cell energetic state. We evaluated the activity of PQQ4R using three *E. coli* isogenic strains differing only in the level of expression of its major efflux pump system— the AcrAB-TolC system - and compared its activity with that of the known *E. coli* efflux inhibitors PAβN and CPZ. We found that PQQ4R interferes with *E. coli* membrane integrity, therefore reducing the activity of *E. coli* efflux pumps. The understanding gathered on the molecular mechanism of action of the efflux inhibitor PQQ4R may aid in the development of less toxic and more potent efflux inhibitors against *E. coli* efflux pumps.

## MATERIALS & METHODS

### Bacteria and growth conditions

The strains included in the study were the wild-type *E. coli* K-12 AG100 (*argE3 thi-1 rpsL xyl mtl*Δ (*gal-uvrB*)*supE44*); the AcrAB pump-deficient *E. coli* AG100A (Δ*acrAB*), and the AcrAB overexpressing *E. coli* AG100$_{tet}$. The strain AG100A is a derivative of AG100 and has the AcrAB system inactivated due to an insertion of the transposon Tn*903* (Δ*acrAB*::Tn*903* Kan$^r$) (*George & Levy, 1983*; *Okusu, Ma & Nikaido, 1996*). AG100$_{tet}$ is a derivative of AG100 obtained by continuous exposure to increasing concentrations of tetracycline (TET) (*Viveiros et al., 2005*). The strains were grown in Luria-Bertani (LB) broth at 37 °C with shaking. AG100A was grown in presence kanamycin (KAN) at 100 µg/ml to maintain the transposon and AG100$_{tet}$ was grown in media supplemented with TET at 8 µg/ml to maintain the overexpression of efflux pumps.

### Chemicals

Ofloxacin (OFX), oxacillin (OXA), KAN, TET, ethidium bromide (EtBr), CPZ, PAβN, carbonyl cyanide-*m*-chlorophenylhydrazone (CCCP), glucose, and phosphate-buffered saline (PBS) were purchased from Sigma-Aldrich (St. Louis, MO, USA). The 2-phenylquinoline PQQ4R was synthesized as previously described (*Sabatini et al., 2013*).

### Antibacterial activity evaluation

The minimum inhibitory concentrations (MICs) were determined using the broth microdilution method according to the CLSI guidelines (*Clinical and Laboratory Standards Institute, 2014*). The synergistic activity between the efflux inhibitors, antibiotics and EtBr was evaluated by checkerboard assays as previously described (*Pillai, Moellering & Eliopoulos, 2005*; *Coelho et al., 2015*). Two-fold serial dilutions of the compounds were made to achieve the following concentrations: 320 µM–40 µM. The MICs were determined as the lowest concentration at which no visible growth was observed after 18 h of incubation at 37 °C. The assays were performed in triplicate and the MIC value was given as result of at least two concordant values.

### Evaluation of efflux activity by real-time fluorometry

The effect of the inhibitors on EtBr accumulation and efflux was assessed by fluorometry as previously described (*Viveiros et al., 2008*; *Viveiros et al., 2010*). The strains were grown until an OD$_{600\ nm}$ of 0.6 at 37 °C with shaking. After, the cells were collected by centrifugation at 16,060× g for 3 min, washed in PBS and centrifuged again.

For the accumulation assays, the OD$_{600\ nm}$ of the cell suspension was adjusted to 0.6 by adding PBS, allowing the assay to run with a final OD of 0.3. To determine the EtBr concentration at which influx and efflux are in equilibrium, the accumulation assays were performed in presence of increasing concentrations of the dye. The assays were prepared to a final volume of 100 µl containing 50 µl of the cellular suspension (final OD$_{600\ nm}$ of 0.3) plus 50 µl of EtBr solutions to final concentrations ranging from 0.0625 to 5 µg/ml. The effect of the inhibitors on the accumulation of EtBr was evaluated in a final volume of 100 µl containing 50 µl of the cellular suspension (final OD$_{600\ nm}$ of 0.3) and 50 µl

**Table 1 Minimum inhibitory concentration (MIC) determination of the antibiotics, EtBr and efflux inhibitors against the *E. coli* strains.**

| | MIC for the *E. coli* strains | | | | | |
| | AG100[a] | | AG100A[b] | | AG100$_{tet}$[c] | |
| Compound | (µg/ml) | (µM) | (µg/ml) | (µM) | (µg/ml) | (µM) |
|---|---|---|---|---|---|---|
| **Antibiotics** | | | | | | |
| OFX | 0.25 | – | 0.0313 | – | 1 | – |
| TET | 2 | – | 0.5 | – | 64 | – |
| OXA | 512 | – | 2 | – | >2,048 | – |
| **Efflux inhibitors** | | | | | | |
| PQQ4R | 256 | 653.9 | 32 | 81.7 | >256 | >653.9 |
| CPZ | 120 | 337.8 | 60 | 168.9 | 280 | 788 |
| PAβN | >200 | >385 | 50 | 96.3 | >200 | >385 |
| **Efflux substrate** | | | | | | |
| EtBr | 200 | – | 3.125 | – | 512 | – |

Notes.

[a] *E. coli* AG100—wild-type.

[b] *E. coli* AG100A—AG100 with the AcrAB-TolC efflux pump inactivated.

[c] *E. coli* AG100$_{tet}$—AG100 with efflux pump overexpression.

CPZ, chlorpromazine; EtBr, ethidium bromide; OFX, ofloxacin; OXA, oxacillin; PAβN, phe-arg- β-naphthylamide; TET, tetracycline.

of a solution containing EtBr at the equilibrium concentration and the compounds to a final concentration of 80 µM for AG100 and AG100$_{tet}$, and 20 µM for AG100A. The molar concentrations used represent $\frac{1}{4}$ or less of the MIC determined for each compound against each strain (see Table 1) in order to guarantee that the real-time efflux inhibitory effects measured were not due to any antimicrobial effect of the compound. The assays were conducted in a Rotor-Gene 3000 (Corbett Research, Sydney, Australia) at 37 °C, and the fluorescence acquired at 530/585 nm at the end of every 60 s, for 30 min. The activity of the compounds on the accumulation of EtBr was evaluated by the relative final fluorescence (RFF) index according to the formula: $RFF = (RF_{treated} - RF_{untreated})/RF_{untreated}$. In this formulae the $RF_{treated}$ corresponds to the fluorescence at the last time point of the EtBr accumulation curve (minute 30) in the presence of an inhibitor and the $RF_{untreated}$ corresponds to the fluorescence at the last time point of the EtBr accumulation curve of the control tube (*Machado et al., 2011*). The experiments were done in triplicate and the RFF values are presented as the average of three independent assays (±SD).

For the efflux assays, the strains were exposed to conditions that promote maximum accumulation of EtBr, i.e., EtBr at the equilibrium concentration for each strain, no glucose, presence of the efflux inhibitors, and incubation at room temperature for 1 h (*Viveiros et al., 2010*). The OD$_{600\ nm}$ of the cell suspension was adjusted to 0.3 and incubated with EtBr under the conditions described above. Aliquots of 50 µl of the cells were transferred to tubes containing 50 µl of each efflux inhibitor at 80 µM (AG100 and AG100$_{tet}$) or 20 µM (AG100A) without EtBr. Control tubes with only cells and cells with and without 0.4% glucose were included. The fluorescence was measured in a Rotor-Gene 3000 and the data was acquired every 30 s for 30 min at 37 °C. The efflux activity was quantified by

comparing the fluorescence data obtained under conditions that promote efflux (presence of glucose and absence of efflux inhibitor) with the data from the control in which the bacteria are under conditions of no efflux (presence of an inhibitor and no energy source). The relative fluorescence corresponds to the ratio of the fluorescence that remains per unit of time, relatively to the EtBr-loaded cells (*Viveiros et al., 2008*; *Viveiros et al., 2010*).

## Time-kill kinetics

The determination of the killing activity of PQQ4R was analysed by time-kill assays as previously described (*Pillai, Moellering & Eliopoulos, 2005*) with slight modifications. Briefly, AG100 was grown until $OD_{600 \, nm}$ of 0.6 at 37 °C with shaking. Exponential bacterial cultures were diluted in Mueller-Hinton Broth (MHB) to a cell density of $1 \times 10^5$ cells/ml and 500 µl added to test tubes containing MHB and PQQ4R at the desired concentration. The compound was added to each tube to achieve the final concentrations of $8\times$ to $0.5\times$ the MIC. A drug-free control was included in the assay to monitor the normal growth of the strain. Cultures were sampled for CFU determination after 0, 1, 2, 3, 4, 5, 6, and 24 h of incubation at 37 °C with shaking. For CFU determination, 10-fold serial dilutions were made in a saline solution and 20 µl of each solution was placed onto the surface of a Mueller–Hinton agar plate. The colonies were counted after the incubation of the plates at 37 °C for 24 h. The limit of detection of the assay was 17 CFU/ml. Each assay was repeated at least twice.

## Membrane potential assay

The effect of PQQ4R on the membrane potential was measured using the BacLight Bacterial Membrane Potential Kit (Molecular Probes, Life Technologies) according to the manufacturer's instructions. CPZ and PAβN were tested for comparison. AG100 was grown at 37 °C with shaking until reach $OD_{600 \, nm}$ 0.6. After this, the cells were washed in PBS and diluted to $1 \times 10^7$ CFU/ml with PBS. PQQ4R was added to the cell suspension at concentrations from 640 to 80 µM. The samples were transferred into black flat bottom 96-well plates, 30 µM of $DiOC_2$ (3) was added to the mixture, and the plates were incubated during 30 min in the dark. The fluorescence was measured using a Synergy HT multi-mode microplate reader (BioTek Instruments Inc, Vermont, USA) with the filters 485/20 (excitation) and 528/20 (emission) for green and 590/35 (emission) for red. CCCP was used as positive control at 156 µM (half MIC), since it eradicates the proton gradient, eliminating the membrane potential. The red to green ratio was determined and normalized against the emission from the $DiOC_2$ (3) blank well and the results are presented as the percentage of depolarized membranes ($\pm$SD) compared with the drug-free control.

## Membrane permeability assay

The evaluation of the membrane integrity was done using the Live/Dead BacLight Bacterial Viability Kit (Molecular Probes, Life Technologies) according to the manufacturer's instructions. AG100 was grown at 37 °C with shaking until $OD_{600 \, nm}$ of 0.6. Samples of 500 µl were incubated with PQQ4R at concentrations from 640 to 80 µM during 1 h at room temperature. The cells were collected by centrifugation at $16,060\times$ g for 10 min, the supernatant was discarded and the pellet resuspended in the same volume of a saline

solution. Then, 100 μl of the cell suspension were transferred into black flat bottom 96-well plates and the dyes propidium iodide and SYTO 9 (ratio 1:1) were added to each well to stain the cells. The plate was incubated during 15 min at room temperature in the dark. The fluorescence was measured using a Synergy HT multi-mode microplate reader with the filters 485/20 (excitation) and 528/20 (emission) for green and 590/35 (emission) for red. The green to red ratio was determined and the results are presented as the percentage of intact membranes (±SD) compared with the control (no treatment). CPZ and PAβN were included for comparison.

## Intracellular ATP levels determination

The ATP levels were measured using the ATP Determination Kit (Invitrogen, Life Technologies, Paisley, UK) according to the manufacturer's instructions. AG100 was grown until $OD_{600\ nm}$ of 0.6 at 37 °C with shaking. Exponential bacterial cultures were diluted in MHB to a cell density of $1 \times 10^5$ cells/ml and 500 μl added to test tubes containing MHB and PQQ4R at half MIC and at the MIC. A drug-free control was included in the assay to monitor the normal growth of the strain. Aliquots of bacteria were collected at 0, 1, 2, 3, 4, 5, 6, and 24 h of incubation at 37 °C, with shaking, inactivated by heating and immediately deep frozen. The cell lysates were transferred into white flat bottom 96-well plates and the ATP content measured using a Synergy HT multi-mode microplate reader and expressed as relative luminescence units. CPZ and PAβN were included for comparison.

## Cytotoxicity against human monocyte-derived macrophages

The blood was collected from healthy donors and the peripheral blood mononuclear cells were isolated by Ficoll-Paque Plus (GE Healthcare, Freiburg, Germany) density gradient centrifugation as previously described (*Machado et al., 2016*). Briefly, monocytes were differentiated into macrophages during 7 days in RPMI-1640 medium with 10% fetal calf serum (FCS), 1% GlutaMAX$^{TM}$, 1 mM sodium pyruvate, 10 mM HEPES at pH 7.4, 100 IU/ml penicillin and 100 μg/ml streptomycin (Gibco, Life Technologies), and 20 ng/ml M-CSF (Immunotools, Friesoythe, Germany) and incubated at 37 °C in a 5% $CO_2$ atmosphere. Fresh medium was added at day 4 post isolation. The effect of the compounds on human monocyte-derived macrophages was evaluated using the AlamarBlue method (Molecular Probes, Life Technologies) (*O'Brien et al., 2000*), and by measuring the release of the lactate dehydrogenase (LDH) into the culture supernatants using the Pierce LDH Cytotoxicity Assay (ThermoFisher Scientific, Waltham, MA, USA) (*Chan, Moriwaki & Rosa, 2013*) according to the manufacturer's instructions. Briefly, $5 \times 10^4$ cells were seeded in 96-well microplates treated with the compounds, and incubated at 37 °C with 5% $CO_2$. After 72 h of treatment, the cell viability was assessed. For the AlamarBlue method, the dye was added to each well to a final concentration of 10% and incubated overnight at 37 °C and 5% $CO_2$. The fluorescence was measured with a 540/35 excitation filter and a 590/20 emission filter in a Synergy HT multi-mode microplate reader. The release of LDH from damaged cells into culture supernatants, as an indicator of cytotoxicity, was measured in serum-free medium. Here the amount of enzyme activity correlates to the number of damaged cells. The specific lysis was calculated as follows: (treated cells—spontaneous

LDH release)/ (maximum LDH release—spontaneous LDH release) ×100. The inhibitory concentration $IC_{50}$ (50%) value was calculated using GraphPad Prism V5.01 software (La Jolla, USA). The $IC_{10}$ (10%) and $IC_{90}$ (90%) were calculated using the GraphPad program "ECanything" available online (*Graph Pad Software, 2016*).

### Statistical analysis

Statistical analysis was carried out using the Student's *t*-test. A *P* value <0.05 was considered statistically significant and highly significant when **$P < 0.01$ and ***$P < 0.001$ (two-tailed tested).

## RESULTS

### Synergistic activity of PQQ4R in combination with antimicrobials

The MICs of the antibiotics, which are substrates of the AcrAB efflux pump, the efflux substrate EtBr, PQQ4R, and for comparison, PAβN and CPZ, known *E. coli* efflux inhibitors, were determined for the wild-type AG100, the pump deficient AG100A, and the pump-overexpressing AG100$_{tet}$ (Table 1). AG100A is more susceptible to the selected antibiotics and EtBr than the wild-type, due to the absence of a functional AcrAB, reconfirming that these antibiotics are substrates of this pump (*Piddock, 2006*; *Viveiros et al., 2005*). The susceptibility data also showed that neither PQQ4R nor PAβN and CPZ had, by themselves, antimicrobial activity against the *E. coli* strains. Moreover, PQQ4R MIC decreased 8-fold against AG100A once compared with the wild-type parental strain, indicating that, PQQ4R might be a substrate of the AcrAB system. Similar results were observed for PAβN.

To assess whether PQQ4R potentiated the activity of the tested antibiotics against *E. coli*, the MICs of OFX, OXA, and TET were determined in the presence of PQQ4R. PAβN and CPZ were tested at the same concentrations for a direct comparison of their activities (Table 2). Against the wild-type strain, PQQ4R, at concentrations ranging from 40 to 160 μM, produced a 2-fold reduction in OFX and TET MICs, while a 2- or 4-fold reduction on the MICs of EtBr at 80 and 160 μM, respectively, were observed. On the contrary, no effect was observed for OXA. Interestingly, PQQ4R at concentrations of 80 and 160 μM, significantly increased the antibacterial activity of OFX and TET in the AcrAB-overexpressing strain AG100$_{tet}$, (AcrAB overexpressed 6–10 times compared with the isogenic wild-type strain—*Viveiros et al., 2005*; *Viveiros et al., 2007*) causing a 4-fold decrease in their MICs, with a marginal effect on the MIC of EtBr. In accordance with the increased susceptibility detected above, the pump deficient AG100A did not grow in presence of PQQ4R at concentrations above 40 μM and this concentration had no effect on the MIC of the antibiotics. However, this concentration caused a ≈8-fold reduction in the EtBr MIC for AG100A, confirming that other efflux pumps, that extrude EtBr (AcrEF, among others), are active in this strain (*Viveiros et al., 2005*; *Viveiros et al., 2007*).

The presence of PAβN significantly reduced the MICs of OXA and TET on the three strains. PAβN reduced the MIC of TET and OXA on the pump-deficient strain by 16-fold and 4-fold, respectively; however, it showed no effect on OFX and EtBr MICs. Conversely, the exposure to PAβN produced a 128-fold decreased in the MIC of EtBr against the

Machado et al. (2017), *PeerJ*, DOI 10.7717/peerj.3168

**Table 2** Synergistic effect of PQQ4R, PAβN, and CPZ on the MIC values of antibiotics and EtBr against the *E. coli* strains.

| | MIC (µg/ml) | | | | | | | | | | | | | | | |
| | OFX | | | | OXA | | | | TET | | | | EtBr | | | |
| **Strain** | **[PQQ4R] (µM)** | | | | **[PQQ4R] (µM)** | | | | **[PQQ4R] (µM)** | | | | **[PQQ4R] (µM)** | | | |
| | 0 | 40 | 80 | 160 | 0 | 40 | 80 | 160 | 0 | 40 | 80 | 160 | 0 | 40 | 80 | 160 |
| AG100 | 0.25 | 0.125 | 0.125 | 0.125 | 512 | 512 | 512 | 512 | 2 | 1 | 1 | 1 | 200 | 200 | 100 | **50** |
| AG100A | 0.03 | 0.03 | – | – | 2 | 1 | – | – | 0.5 | 0.25 | – | – | 3.125 | **0.39** | – | – |
| AG100 tet | 1 | 0.5 | **0.25** | **0.25** | >2,048 | >2,048 | 2,048 | 2,048 | 64 | **16** | **16** | **16** | 512 | 512 | 512 | 512 |
| | **[PAβN] (µM)** | | | | **[PAβN] (µM)** | | | | **[PAβN] (µM)** | | | | **[PAβN] (µM)** | | | |
| | 0 | 40 | 80 | 160 | 0 | 40 | 80 | 160 | 0 | 40 | 80 | 160 | 0 | 40 | 80 | 160 |
| AG100 | 0.25 | **0.06** | **0.06** | **0.06** | 512 | **64** | **64** | **64** | 2 | 2 | 1 | **0.06** | 200 | 100 | 100 | 100 |
| AG100A | 0.03 | 0.03 | – | – | 2 | **0.25** | – | – | 0.5 | **0.03** | – | – | 3.125 | 3.125 | – | – |
| AG100tet | 1 | **0.06** | **0.03** | **0.03** | >2,048 | **256** | **256** | **64** | 64 | **8** | **4** | **2** | 512 | 256 | **128** | **4** |
| | **[CPZ] (µM)** | | | | **[CPZ] (µM)** | | | | **[CPZ] (µM)** | | | | **[CPZ] (µM)** | | | |
| | 0 | 40 | 80 | 160 | 0 | 40 | 80 | 160 | 0 | 40 | 80 | 160 | 0 | 40 | 80 | 160 |
| AG100 | 0.25 | 0.125 | 0.125 | **0.003** | 512 | 512 | 512 | **≤1** | 2 | 1 | **0.5** | **≤0.03** | 200 | 100 | **25** | **≤0.39** |
| AG100A | 0.03 | 0.015 | – | – | 2 | 2 | – | – | 0.5 | **0.125** | – | – | 3.125 | **0.39** | – | – |
| AG100 tet | 1 | 0.5 | 0.5 | **0.25** | >2,048 | >2,048 | 2,048 | **1,024** | 64 | 32 | 32 | **16** | 512 | 256 | **128** | **128** |

**Notes.**

[a] *E. coli* AG100—wild-type.

[b] *E. coli* AG100A—AG100 with the AcrAB-TolC efflux pump inactivated.

[c] *E. coli* AG100tet—AG100 with efflux pump overexpression

CPZ, chlorpromazine; EtBr, ethidium bromide; OFX, ofloxacin; OXA, oxacillin; PA βN, phe-arg-β-naphthylamide; TET, tetracycline; –, AG100A does not grow at concentration above 40 µM of PQQ4R.

**Table 3  Relative final fluorescence values (RFF) based on the accumulation of EtBr for the *E. coli* strains in the presence of the efflux inhibitors.**

| AG100[a] | | AG100A[b] | | AG100$_{tet}$[c] | |
|---|---|---|---|---|---|
| Compound (80 µM) | RFF | Compound (20 µM) | RFF | Compound (80 µM) | RFF |
| PQQ4R | **13.1 ± 0.04\*\*\*** | PQQ4R | **1.5 ± 0.1\*\*** | PQQ4R | **5.2 ±0.9\*\*\*** |
| CPZ | **8.1 ± 1.1\*\*\*** | CPZ | **2.0 ± 0.04\*\*** | CPZ | **7.9 ±0.4\*\*\*** |
| PAβN | 0.7 ± 1.2\* | PAβN | 0.1 ± 0.1 | PAβN | **1.3 ± 0.3\*\*** |

**Notes.**
[a] *E. coli* AG100—wild-type.
[b] *E. coli* AG100A—AG100 with the AcrAB-TolC efflux pump inactivated.
[c] *E. coli* AG100$_{tet}$—AG100 with efflux pump overexpression.
CPZ, chlorpromazine; EtBr, ethidium bromide; PAβN, phe-arg- β-naphthylamide; Accumulation of EtBr at 1 µg/ml (AG100), 0.25 µg/ml (AG100A) and 2 µg/ml (AG100$_{tet}$) in the absence of glucose; Values in bold type (RFF ≥1) indicated enhanced accumulation of EtBr in presence of an efflux inhibitor. The results presented correspond to the average of three independent assays plus standard deviation (±SD). Results were considered significant when \*$P < 0.05$ and highly significant when \*\*$P < 0.01$ and \*\*\*$P < 0.001$; CPZ, chlorpromazine; PAβN, phe-arg- β-naphthylamide, EtBr, ethidium bromide.

AcrAB-overexpressing strain, while no change was observed in the MIC of the wild-type strain. The inability of PAβN to reduce the MIC of EtBr in wild-type strains has already been described for *E. coli* and *P. aeruginosa* (*Lomovskaya et al., 2001*; *Kern et al., 2006*). This result indicates that PAβN is acting as a pump competitor and not as an efflux inhibitor like PQQ4R and CPZ. The activity of CPZ against the pump-deficient strain was similar to that of PQQ4R, with the exception that CPZ reduced the MIC of TET by 4-fold. CPZ significantly reduced the MICs of OFX, OXA, TET and EtBr on the wild-type and the AcrAB-overexpressing strain.

## Efflux inhibitory activity of PQQ4R

The MIC results give an indirect measure of the potential efflux activity of the strains and the effect of the inhibitors on this activity. The ability of PQQ4R to inhibit *E. coli* efflux systems was confirmed by real-time fluorometry using EtBr, a broad efflux pump substrate (*Viveiros et al., 2008*; *Viveiros et al., 2010*). Initially, it was determined the equilibrium concentration at which the influx of EtBr equals its efflux. The accumulation of EtBr started at concentrations above 1 µg/ml for AG100, 0.25 µg/ml for AG100A, and 2 µg/ml for AG100$_{tet}$ (Fig. S1). Using these concentrations, we evaluated the ability of PQQ4R, at sub-inhibitory concentrations (<1/8 MIC), to promote the intracellular accumulation of EtBr (Fig. 1) and calculated the RFFs (Table 3). The RFF value is a measure of how effective the compound is on the inhibition of the EtBr efflux (at a given concentration) by comparison of the final fluorescence at the last time point (60 min) of the treated cells with the cells treated only with EtBr. An index of activity above zero indicated that the cells accumulate more EtBr under the condition used than those of the control (non-treated cells taken as 0). In case of negative RFF values, these indicated that the treated cells accumulated less EtBr than those of the control condition. Values above 1 in the presence of the efflux inhibitors were considered enhanced accumulation of EtBr inside the cells. CPZ and PAβN were included at the same concentrations for a direct comparison of their inhibitory potencies.

At 80 µM, PQQ4R showed an RFF of 13.1 against the wild-type strain, indicating a strong ability to interfere with EtBr efflux compared to CPZ and PAβN, that showed RFF values of 8.1 and 0.7, respectively (Fig. 1A; Table 3). This means that PQQ4R promotes the higher

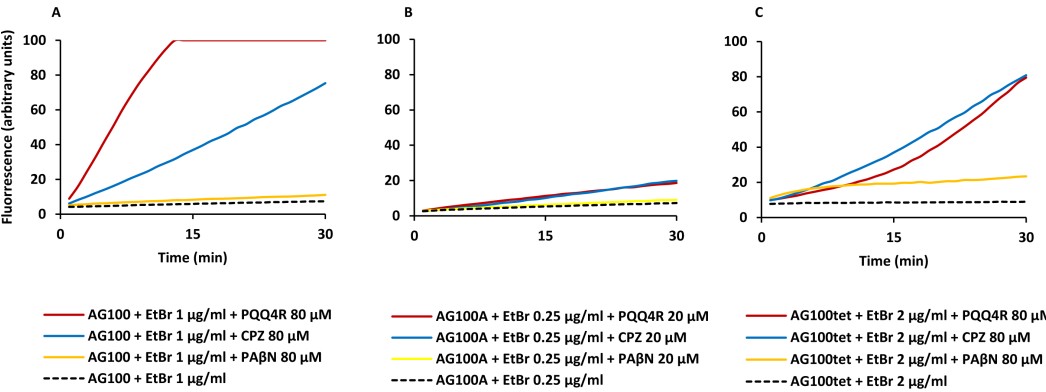

**Figure 1** **Effect of PQQ4R, CPZ, and PAβN on the accumulation of ethidium bromide by *E. coli*.** (A) AG100; (B) AG100A; and (C) AG100<sub>tet</sub>. The equilibrium concentration of ethidium bromide was 1 µg/ml for AG100, 0.25 µg/ml for AG100A, and 2 µg/ml for AG100<sub>tet</sub>. Assays were performed at 37 °C in the absence of glucose. The inhibitors were used at 80 µM for AG100 and AG100<sub>tet</sub>, and 20 µM for AG100A. EtBr, ethidium bromide; CPZ, chlorpromazine; PAβN, phenyl-arginine- β-naphthylamide.

accumulation levels of EtBr compared to CPZ followed by PAβN. In the AcrAB-deficient strain there was a small increase in the accumulation of EtBr in the presence of PQQ4R (RFF = 1.5) and CPZ (RFF = 2), whereas almost no effect was observed with PAβN (RFF = 0.1) (Fig. 1B; Table 3). These results showed once again that AcrAB is the main efflux system that pumps out EtBr in *E. coli*, but that other efflux pumps susceptible to PQQ4R and CPZ are pumping out EtBr as described before (*Viveiros et al., 2005*; *Viveiros et al., 2007*). For the AcrAB-overexpressing strain AG100<sub>tet</sub> (AcrAB is 6–10 times overexpressed compared to the wild-type strain), the accumulation of EtBr in the presence of PQQ4R corresponded to a RFF of 5.2 (Fig. 1C; Table 3). The EtBr accumulation in this strain increased significantly when compared to AG100A, but, for the same molar concentration of inhibitor tested (80 µM), did not reach the same levels of inhibition observed for AG100, since the AcrAB system is 6–10 times more expressed in the AG100<sub>tet</sub> strain compared with the AG100 (*Viveiros et al., 2008*; *Viveiros et al., 2010*). These results, obtained in a set of three isogenic *E. coli* that differ only by the absence/presence or expression level of the AcrAB system, indicated that PQQ4R promotes accumulation (i.e., interferes with the efflux) of EtBr in *E. coli* mainly thought the inhibition of the AcrAB efflux system. In the presence of CPZ and PAβN, the EtBr accumulation in the AG100<sub>tet</sub> reaches similar levels (RFF 7.9 and 1.3, respectively) to those obtained for the wild-type strain (RFF 8.1 and 0.7, respectively), being less effective in the inhibition of efflux activity than PQQ4R.

To confirm the results obtained above on the accumulation of EtBr promoted by the inhibitors, we performed efflux assays for each strain. Each strain was subject to conditions that promote significant EtBr accumulation over a period of 60 min at room temperature: in the absence of glucose and presence of PQQ4R, CPZ or PAβN. After the maximum accumulation has been reached, EtBr and the efflux inhibitors were washed-out and the cells were subsequently re-suspended in new buffer with and without glucose and the inhibitor. As showed in Fig. 2, the efflux take place readily in presence of glucose at 37 °C, an activity that is inhibited in the presence of PQQ4R and CPZ and to a lesser extent with

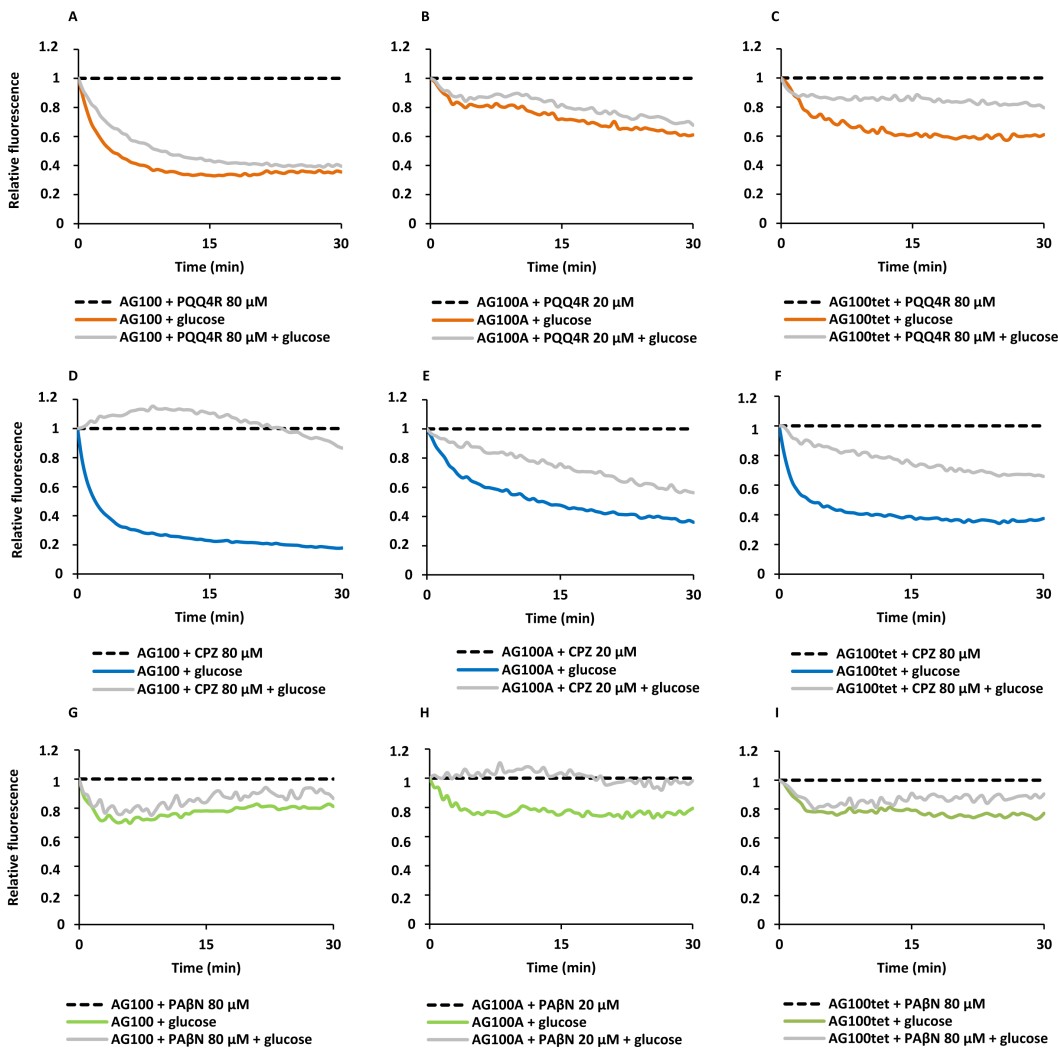

**Figure 2 Effect of PQQ4R, CPZ, and PAβN on the efflux of EtBr by the *E. coli* strains.** (A) AG100;
(B) AG100A; and (C) AG100tet. The assays were performed at 37 °C in the presence and absence of glu-
cose. The concentrations of PQQ4R, CPZ and PAβN were 80 μM for AG100 and AG100tet, and 20 μM for
AG100A. EtBr, ethidium bromide; CPZ, chlorpromazine; PAβN, phenyl-arginine-β-naphthylamide.

PAβN. These results confirmed that PQQ4R, CPZ, and PAβN inhibit the efflux of EtBr
and this inhibitory effect is transient when the cells are washed-out of the inhibitor and an
energy source is given to promote active efflux reenergizing the cells (Fig. 2).

## Membrane depolarization and cell viability

To test the hypothesis that the interference with the bacterial energy metabolism is the
cause of the efflux inhibition by PQQ4R, the effect of this compound on the bacterial
membrane potential was evaluated using the BacLight Bacterial Membrane Potential
Kit. Thirteen minutes after adding PQQ4R at any of the concentrations tested, the
percentage of depolarized cells was over 85% (Fig. 3A). The protonophore CCCP was used
as depolarization control and, as expected, the membrane potential collapsed when the cells
were incubated for the same time with CCCP (over 95%). These results showed that PQQ4R

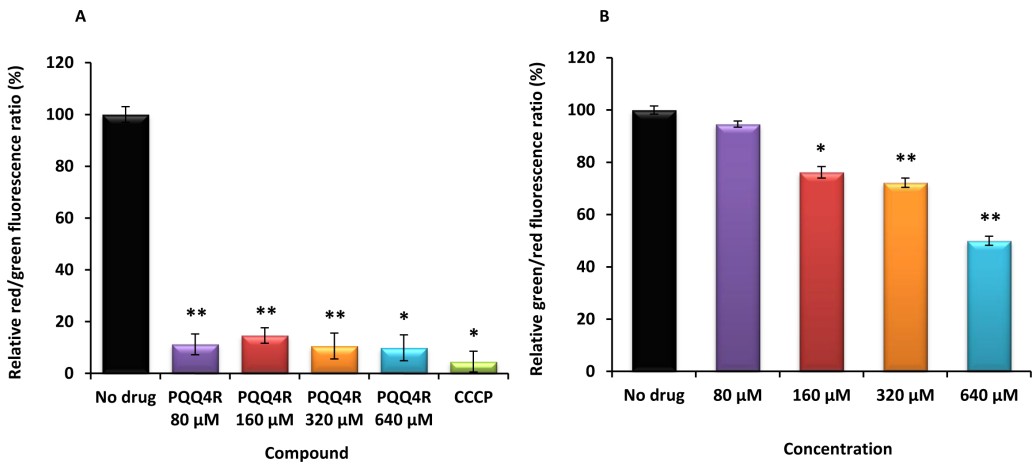

**Figure 3** **Membrane depolarization (A) and permeabilization (B).** The relative red/green ratio of *E. coli* using DIOC$_2$ (3) stained cells after 30 min of exposure to PQQ4R from 80 µM to 640 µM. Green fluorescence corresponds to the depolarized cells; red fluorescence corresponds to the polarized cells. The changes in the fluorescence were measured at an excitation wavelength of 485/20 nm and 528/20 (emission) for green and 590/35 (emission) for red; (B) Membrane permeability measured with Baclight Live/Dead assay upon exposure to PQQ4R. Green fluorescence corresponds to the cells with intact membranes; red fluorescence corresponds to cells with permeabilized membranes cells. The data was normalized against the drug-free control. CCCP was used as positive control, at 156 µM (half MIC), since it eradicates the proton gradient, eliminating membrane potential. The results presented correspond to the average of two independent assays plus standard deviation (±SD). Results were considered significant when $^*P < 0.05$ and highly significant when $^{**}P < 0.01$ and $^{***}P < 0.001$.

depolarizes *E. coli* membranes by interfering with the gradient of protons through the cell membrane. Then, to evaluate if the depolarization of the cells, promoted by PQQ4R, results on a transient effect on cells with an intact membrane or alters the membrane integrity, the Live/Dead BacLight Bacterial Viability Kit was used. This assay uses the fluorescent stain SYTO 9 that penetrates in all bacterial membranes and stains the cells in green, and propidium iodide that penetrates only in the cells with permeabilized membranes. The combination of the two stains produces red fluorescent cells and these latter ones are considered dead by cell damage. The integrity of *E. coli* membranes was evaluated, by fluorometry, after exposure to PQQ4R at sub-inhibitory (1/4 and 1/2 MIC) and bactericidal concentrations (MIC), during 1 h. The results showed that at half MIC, PQQ4R caused cell damage, reducing the membrane integrity by 28% (Fig. 3B). This increased membrane permeability was accompanied by a reduction in the bacterial viability of approx. 1 log (Fig. 4—see also kinetics of bacterial killing). Membrane integrity was then evaluated on the cells exposed to PQQ4R at 80 µM (≈1/8 MIC), the same concentration used in the real-time fluorometry accumulation and efflux assays, for 1 h and the results were assessed as described before. At 80 µM, PQQ4R did not alter the cell membrane integrity (Fig. 3B). To evaluate if the EtBr accumulation observed in presence of CPZ and PAβN was due to membrane permeability we tested both compounds also at 80 µM. The results showed that the membrane permeability increased by 18% and 14% with CPZ and PAβN, respectively. Membrane depolarization occurs in presence of CPZ at 80 µM similarly to that caused by PQQ4R, and PAβN induced 14% membrane depolarization at 80 µM. Overall, these
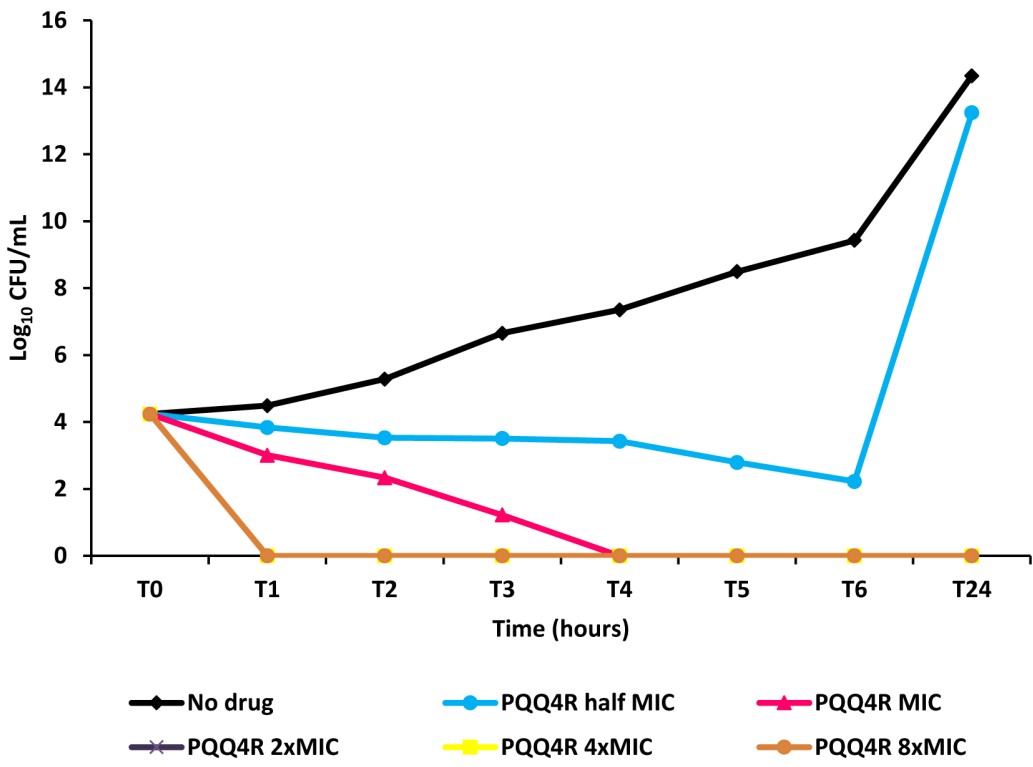

**Figure 4  Effect for PQQ4R on survival of *E. coli*. AG100.**  The compound was added at concentrations from $0.5\times$ MIC ($320$ μM) to $8\times$ MIC ($5,120$ μM) to exponential growing *E. coli* and survival monitored during 24 h.

results showed that PQQ4R causes transient membrane depolarization without affecting membrane integrity or cell viability at low concentrations (80 μM - 1/8 MIC- cell viability at this concentration 14.34 $\log_{10}$ CFU/ml—Fig. 4). In contrast, and as expected by the previous results, at bactericidal concentrations, PQQ4R acts as a membrane-destabilizing agent, which is also consistent with the cell death at this concentration (Fig. 4). The gathering of these results points towards the depolarization of the cell membrane as the main mechanism for PQQ4R's transient efflux-inhibition activity.

## PQQ4R effect on intracellular ATP levels

Destabilization of the membrane functions can impair the respiratory chain functions and consequently reduce the ATP levels. To evaluate whether the exposure to PQQ4R could have an effect on the intracellular ATP levels, *E. coli* AG100 was exposed to PQQ4R and the intracellular ATP levels measured during 24 h at either sub-inhibitory (320 μM −1/2 MIC) or bactericidal concentrations (640 μM—MIC) (Fig. 5). The ATP levels remained constant during the first 3 h of exposure at both concentrations and at the same level of the drug-free control excluding the abrupt ATP depletion from being the direct cause of the efflux-inhibition previously seen with the EtBr accumulation and efflux assays, and supports a slow but steady ATP production impairment. After 4 h of exposure, the ATP levels of the drug-free control increased but the ATP levels of the drug-containing tubes remained practically constant after the addition of PQQ4R and until the end of the assay,
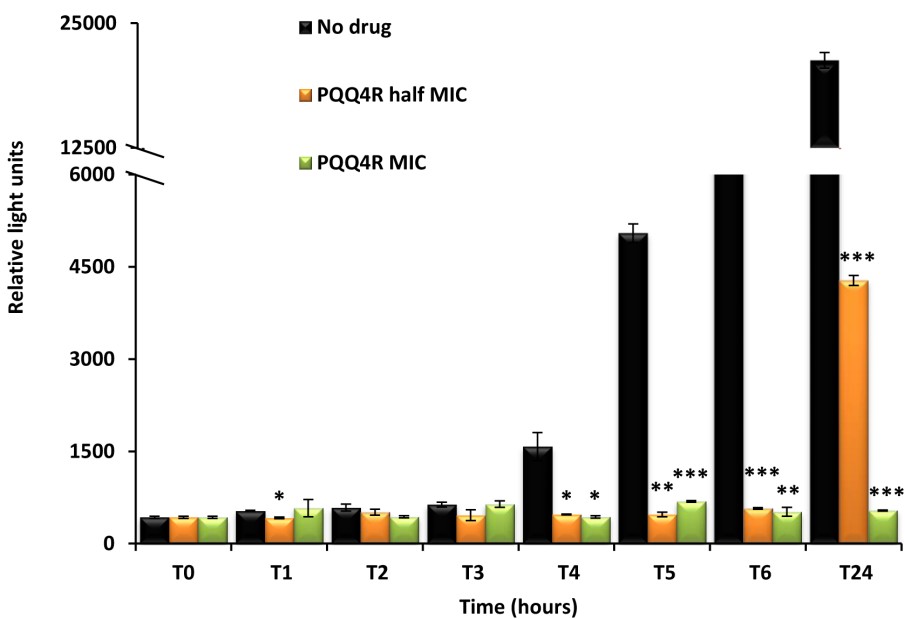

**Figure 5**  **Effect of PQQ4R on *E. coli* ATP levels.** *E. coli* AG100 was exposed to PQQ4R at half MIC and MIC/MCB during 24 h. The ATP levels were quantified using a luciferin-luciferase bioluminescence detection assay as described in 'Material and Methods'. The results presented correspond to the average of two independent assays plus standard deviation ($\pm$SD). Results were considered significant when $^{*}P < 0.05$ and highly significant when $^{**}P < 0.01$ and $^{***}P < 0.001$.

with the exception of the cells exposed to half MIC. In this case, the ATP production increased after 24 h of exposure but was 80% less than that of the drug-free control.

## Kinetics of PQQ4R bacterial killing

To characterize the killing effect of PQQ4R, we measured its bacterial killing activity against AG100 through time-kill studies (Fig. 4). PQQ4R at half MIC had no effect on the viability of *E. coli* after 24 h of exposure. Exposure to PQQ4R reduced the viability of *E. coli* to zero after 3 h at the MIC (256 µg/ml–640 µM) while at 2× MIC PQQ4R reduced the viability to zero after 1 h of exposure. After 24 h of exposure, all cultures remained negative at these concentrations. These results showed that PQQ4R killing activity is rapid reaching 100% lethality at the MIC, a concentration that was also found to be bactericidal (minimal bactericidal concentration—MBC).

## Spectrum of activity of PQQ4R

The antimicrobial activity of PQQ4R was evaluated against other bacterial species to assess its antibacterial specificity: the Gram-negative bacteria *Acinetobacter baumannii*, *Salmonella enterica* serovar Enteritidis, *Klebsiella pneumoniae* and *Enterobacter aerogenes*, the Gram-positive *S. aureus*, and the acid-fast bacteria *Mycobacterium smegmatis*, *M. avium*, and *M. tuberculosis*. The results are depicted in Table 4 and showed that PQQ4R possessed moderate antibacterial activity towards Gram-positive (average MIC 50 µg/ml) and acid-fast bacteria (average MIC 32 µg/ml) and reduced antibacterial activity against Gram-negative bacteria (average MIC $\geq$ 128 µg/ml).

**Table 4  Antibacterial specificity of PQQ4R against a panel of selected bacterial species.**

| Species | Strain | MIC (µg/ml) PQQ4R |
|---|---|---|
| *Enterobacter aerogenes* | ATCC13048 | >256 |
| *Salmonella enterica* serovar Enteritidis | NCTC13349 | 256 |
| *Escherichia coli* K-12 | AG100 | 256 |
| *Acinetobacter baumannii* | ATCC19606[T] | 128 |
| *Klebsiella pneumoniae* | FF4891 | >256 |
| *Staphylococcus aureus* | ATCC25923 | 50 |
| *Mycobacterium smegmatis* mc[2]155 | ATCC700084[T] | 32 |
| *Mycobacterium avium* | 104 | 32 |
| *Mycobacterium tuberculosis* H37Rv | ATCC27294[T] | 32 |

**Table 5  Cytotoxic concentrations (CC) of PQQ4R, PAβN, and CPZ against human monocyte-derived macrophages after three days of exposure.**

| Compound | Cytotoxic concentrations (µM) | | |
|---|---|---|---|
| | CC90 | CC50 | CC10 |
| PQQ4R | 12.73 | 10.83 | 9.21 |
| CPZ | 37.01 | 22.24 | 13.36 |
| PAβN | 3596.86 | 1269 | 447.71 |

**Notes.**

CPZ, chlorpromazine; PAβN, phe-arg-β-naphthylamide.

## Cytotoxicity

Finally, PQQ4R was evaluated for its toxicity against human monocyte-derived macrophages using CPZ and PAβN for comparison. After 72 h of exposure, the viability of the macrophages was assessed using the AlamarBlue method. The comparative activities, $CC_{90}$, $CC_{50}$ and $CC_{10}$ values, are presented in Table 5. The dose–response curves are shown in Fig. 6. PQQ4R showed to be more toxic ($CC_{50}$ 10.83 µM) when compared to CPZ ($CC_{50}$ 22.24 µM) and PAβN ($CC_{50}$ 1269 µM). Afterwards, the amount of LDH release was measured in the cultures supernatants (Fig. 7). The LDH is an intracellular enzyme that is released in the culture media as a consequence of damaged cell membranes (*Chan, Moriwaki & Rosa, 2013*). The treatment with PQQ4R at 0.625 µM–2.5 µM caused 1%–4.6% increase in LDH release compared with the non-treated cells; at 5 µM LDH release increases by 52.45% and to 72%–78% at higher concentrations. Comparatively, CPZ increases LDH release from 0.95% at 1.25 µM to 24% at 20 µM. At concentrations above, the amount of LDH detected in the culture supernatant was above 95%. Concerning PAβN, the amount of LDH detected varied from 1.08% to 8% at concentrations from 10 µM to 640 µM. At concentrations above, PAβN caused 66%–68% LDH release. The results obtained with the LDH method corroborated the results obtained with the AlamarBlue method.

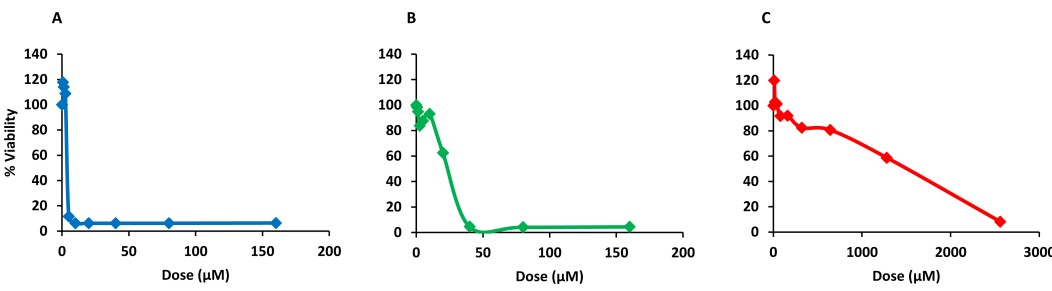

**Figure 6  Dose response curves showing the effect of PQQ4R, CPZ and PAβN against human-monocyte derived macrophages.**  The cells were treated with different concentrations of (A) PQQ4R, (B) CPZ, and (C) PAβN for 3 days; AlamarBlue was then added and cells were further incubated overnight at 37 °C, 5% $CO_2$. The fluorescence was measured Synergy HT multi-mode microplate reader. CPZ, chlorpromazine; PAβN, phenyl-arginine-β-naphthylamide.

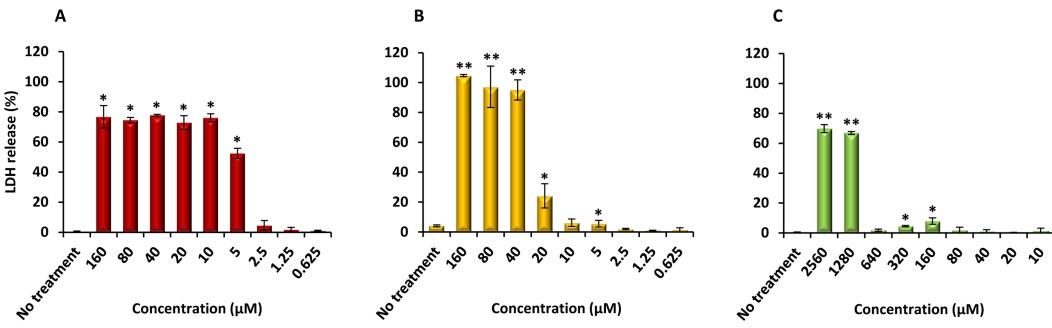

**Figure 7  Effect of PQQ4R, CPZ and PAβN on LDH release from human monocyte-derived macrophages.**  The cells were exposed to each compound during 3 days at 37 °C, 5% $CO_2$. The results presented correspond to the average of at least two independent assays plus standard deviation (±SD). Results were considered significant when $*P < 0.05$ and highly significant when $**P < 0.01$ and $***P < 0.001$.

## DISCUSSION

To detail the biological activity on the 2-phenylquinoline efflux inhibitor PQQ4R against *E. coli*, we investigated first the synergistic activity of PQQ4R in combination with antibiotics that are known substrates of the AcrAB efflux pump. The results showed that PQQ4R acts synergistically, in a concentration dependent manner, with the antibiotics TET and OFX. When compared to CPZ and PAβN, PQQ4R is less potent in its synergistic activity at equivalent molarities. Next, we explored the effect of PQQ4R on the real-time active efflux of the AcrAB efflux substrate EtBr. PQQ4R, CPZ, and PAβN clearly inhibited the efflux of EtBr in a transient manner at sub-inhibitory concentrations (1/8 MIC or less), as shown by the reactivation of active efflux when the cells are released from the inhibitor and reenergized by glucose. As expected, the re-energization effect was more significant with CPZ (known to inhibit AcrAB-mediated EtBr efflux by energy depletion) (*Viveiros et al., 2005*) compared with PAβN, an AcrAB substrate competitor partially affecting membrane integrity (*Misra et al., 2015*). PQQ4R clearly demonstrated a real-time inhibitory effect on efflux similar to CPZ.

In terms of antimicrobial properties, PQQ4R demonstrated a rapid bactericidal activity against *E. coli* with the complete loss of viable cells after only 1 h incubation at $2\times$ MIC and after 4 h at its MIC. This result is consistent with our hypothesis of an alternative mode of action in which at low concentrations PQQ4R inhibit *E. coli* efflux systems and near the MBC it interact directly with *E. coli* membranes. Our results showed that PQQ4R mechanism of action involves the loss of membrane integrity after 1 h of exposure. PQQ4R disrupts both inner and outer membrane, as the uptake of the membrane-impermeant fluorescent dye propidium iodide requires the damage of both to enter in the cytoplasm, to bind to the DNA and fluoresce. Recently, it was showed that PAβN also permeabilizes the *E. coli* outer membrane (*Misra et al., 2015*) and both outer and inner membranes of *P. aeruginosa* (*Lamers, Cavallari & Burrows, 2013*).

The loss of the membrane integrity is closely related with the loss of the cell capacity to synthesize ATP, decreased protein synthesis and to inhibit respiration (*Brogden, 2005*). Our data showed a moderate increase in the ATP levels after the addition of PQQ4R at sub-inhibitory concentrations (half MIC), associated with a decrease in the membrane permeability. At bactericidal concentrations (MIC), the ATP levels decreased and were accompanied by an increase in the membrane permeability. To further support these results, and based upon the fact that the RND efflux pumps act using the electrochemical gradient generated by the PMF (*Anes et al., 2015*), we assessed whether treatment with PQQ4R could have a direct effect on membrane depolarization. Our results showed that PQQ4R causes a significant, but non-lethal, inner membrane depolarization at sub-inhibitory concentrations. At bactericidal concentrations, the membrane depolarization is accompanied by an impaired membrane integrity and ATP production, with cell death. The sequence of events describing PQQ4R mechanism of action is depicted in Fig. 8. Briefly, PQQ4R at low concentrations inhibit *E. coli* efflux systems by interfering with the energy necessary to maintain the pumps working, i.e., PQQ4R is an energetic inhibitor like CPZ and inhibits the efflux activity through the transient dissipation of the membrane potential without affecting the membrane permeability. Also similar to CPZ and PAβN, at higher concentrations PQQ4R permeabilizes *E. coli* membranes causing irreversible cell damage leading to the cell death. Of note, despite several attempts, we could not obtain *E. coli* mutants to PQQ4R. This result showed that PQQ4R has an off-target effect and corroborates our findings that points to membrane disruption as the primary mechanism of action of this molecule.

Regrettably, PQQ4R revealed to be relatively toxic to the human macrophage, with a toxicity level comparable to that of CPZ, an antipsychotic already used in clinical practice and previously shown to have activity against intracellular *M. tuberculosis* at non-toxic clinical concentrations due to its accumulation inside the phagolysosomal macrophage compartments (*Machado et al., 2016*). In a recent study, we showed that PQQ4R, when used at non-toxic concentrations, is also able to potentiate clarithromycin activity against intracellular *M. avium* in a manner similar to CPZ (*Machado et al., 2015*). On the contrary, PAβN showed almost no toxicity against the human macrophages, however, PAβN and its derivatives proved to be nephrotoxic as result of their accumulation in lysosomes. This adversity has hampered the clinical development of PAβN (*Watkins et al., 2003*).

**Figure 8 Sequence of events describing PQQ4R mechanism of action.** PQQ4R mode of action involves efflux pump inhibition, membrane permeabilization, disruption of membrane potential, and ATP depletion. PQQ4R act as efflux inhibitor at low concentrations (80 μM) by interference with energy required for efflux due to its interference with inner membrane potential. At sub-inhibitory concentrations (160 μM and 320 μM), PQQ4R destabilizes *E. coli* membrane functions in a non-permanent manner as suggested by the reduction of ATP levels, depolarization of membrane potential and increased membrane permeabilization; at the MBC (640 μM), membrane permeabilization increases and ATP is lost.

The intriguing activity of PQQ4R may be used as a starting point for the much-needed medicinal chemistry approaches to identify potent RND efflux pumps inhibitors endowed of a suitable safety profile.

In conclusion, PQQ4R is an excellent hit candidate to be studied in depth to find new RND efflux pumps inhibitors of Gram-negative bacteria that act by interfering with the PMF. It shows a broad usefulness as efflux pump inhibitor and, due to its membrane permeabilizing properties, can be used in combination therapies to assist other molecules to enter the bacterial cell (*Lamers, Cavallari & Burrows, 2013*; *Herbel & Wink, 2016*). Moreover, this compound complies with the conditions of *Farha et al.*'s et al., (*2013*) hypothesis that presented an alternative approach to tackle pathogens by using chemical combinations targeting the PMF. The authors showed that combinations between dissipaters of membrane potential with dissipaters of the transmembrane proton gradient are highly synergistic against methicillin-resistant *S. aureus*. This combination will allow reducing each compound individual dose and consequently, their toxicity (*Farha et al., 2013*). Since drug resistance mediated by efflux pumps depend largely on the PMF, dissipaters of the PMF as PQQ4R could be regarded as putative adjuvants of the conventional therapy against bacterial pathogens. Nevertheless, due to the similarities between the bacterial and the mitochondrial electron transport chain, the effect of PQQ4R on the latter will need to be evaluated in the future. Medicinal chemistry studies can now help to improve PQQ4R scaffold to potentiate its permeabilizing and efflux inhibitory properties and reduce its toxicity towards human cells, contributing for the development of new drugs with potential for the clinical usage as adjuvants of the therapy against drug resistant bacterial pathogens.

### Funding
This work was partially supported by projects PTDC/BIA-MIC/121859/2010 and GHTM-UID/Multi/04413/2013 from Fundação para a Ciência e a Tecnologia (FCT), Portugal. DM, LF and SSC were supported by grants SFRH/BPD/100688/2014, PTDC/BIA-MIC/121859/2010, and SFRH/BPD/97508/2013, respectively, from Fundação para a Ciência e a Tecnologia (FCT), Portugal. There was no additional external funding received for this study. The funders had no role in study design, data collection and analysis, decision to publish, or preparation of the manuscript.

### Grant Disclosures
The following grant information was disclosed by the authors:
Fundação para a Ciência e a Tecnologia (FCT), Portugal: PTDC/BIA-MIC/121859/2010, GHTM-UID/Multi/04413/2013, SFRH/BPD/100688/2014, SFRH/BPD/97508/2013.

### Competing Interests
The authors declare there are no competing interests.

### Author Contributions
- Diana Machado conceived and designed the experiments, performed the experiments, analyzed the data, wrote the paper, prepared figures and/or tables, reviewed drafts of the paper.
- Laura Fernandes performed the experiments, analyzed the data, reviewed drafts of the paper.
- Sofia S. Costa analyzed the data, reviewed drafts of the paper.
- Rolando Cannalire, Giuseppe Manfroni and Oriana Tabarrini analyzed the data, reviewed drafts of the paper, assisted with the synthesis of the PQQ4R compound.
- Isabel Couto analyzed the data, contributed reagents/materials/analysis tools, reviewed drafts of the paper.
- Stefano Sabatini analyzed the data, contributed reagents/materials/analysis tools, reviewed drafts of the paper, planed, designed and directed the synthesis of the PQQ4R compound.
- Miguel Viveiros conceived and designed the experiments, analyzed the data, contributed reagents/materials/analysis tools, wrote the paper, reviewed drafts of the paper.

### Data Availability
The raw data has been supplied as a Supplementary File.

### Supplemental Information
Supplemental information for this article can be found online at http://dx.doi.org/10.7717/peerj.3168#supplemental-information.

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
