# Peer review of "Mode of action of the 2-phenylquinoline efflux inhibitor PQQ4R against Escherichia coli"

_PeerJ, doi:10.7717/peerj.3168_

## Round 0.1 · original submission · Major Revisions

After considering all reviewers comments I think it is necessary to address all concerns raised by reviewers 2 & 3 in order to reconsider the possible acceptance of this manuscript for PeerJ. I understand the work is technically sound although concerns about the possibility of application of PQQ4R because of its apparent toxicity to cell lines used here, along with the number of issues found by reviewer 3, make me reach this conclusion. I sincerely expect a thoroughly revised version of your work soon.

Reviewer 1 ·

Basic reporting

NO comment

Experimental design

No Comment

Validity of the findings

The biggest issue is that PQQ4R is cytotoxic, which casts doubt on all the experimental data generated and its eventual utilization.

Additional comments

The biggest issue is that PQQ4R is cytotoxic, which casts doubt on all the experimental data generated and its eventual utilization.

·

Basic reporting

The manuscript of Machado et al in general is well written.The references are enough for the work. However in the section of cytotoxicity against macrophages, no references are shown. The first section of the manuscript is well presented, however the high toxicity of PQQ4R on macrophages weakens the concept of antimicrobial molecule.

Experimental design

In general, the experimental design is well done. The weakest section is relative to the toxicity on macrophages. The method used (AlarmBlue) involves incubation overnight, so I suggest another assays faster to evaluate the toxicity, i.e. determination of LDH in supernatants or analysis of death cell using Propidium Iodide/Annexin V by flow cytometry.

Validity of the findings

The findings are validated with exception of the toxicity on macrophages.

Additional comments

The work clearly shows the effect of PQQ4R molecule on the viability of E. coli. The authors attempt to propose this molecule as antibiotic for clinical usage against drug resistant bacterial infections even when this molecule is highly toxic to human macrophages. In this hand, I would suggest to focus the study to the molecular mechanisms of PQQ4R (i.e ATP levels, changes in the inner membrane and eflux pumps) on E. coli. To support the observations of this work I propose the following:

1) Analysis of the toxicity of PQQ4R on macrophage by determination of LDH in supernatants or analysis of death cell using Propidium Iodide/Annexin V by flow cytometry.
2) Should indicate the individual MIC for both Gram positive and Gram negative bacteria (the authors show the average). In this point, should be important to evaluate Gram negative bacteria with mucoid phenotype such as Klebsiella pneumoniae or Pseudomonas aeruginosa.

Reviewer 3 ·

Basic reporting

English language: The quality of writing was comprehensible but numerous grammatical issues (omission or improper use of articles, in appropriate adverb/verb tenses, unnecessary adjectives, and word pluralizations) were noted in every paragraph. The authors should consider using a grammatical language editing program or english speaker to complete the revisions.

Intro and background: The introduction provides some details regarding efflux pumps which are appropriately referenced but overall, the introduction lacks important information regarding the efflux pump inhibitors, what their mode of actions are and a clear rationale for why PQQ4R was selected for study. The introduction would benefit by including the missing hypothesis presented in the middle of the results. Some of these concerns are addressed by moving paragraphs from the discussion.

The structure of the paper: appears to conform to the basics outlined by PeerJ.

Figures and Tables: The image quality of the figures is sufficient for publication (the word sizing on Fig 7 may need to be increased for legibility). Almost all figures have commas where decimals should be used. Inconsistent usage of abbreviations are noted in most figures and text. Most figures and tables would benefit from the inclusion of standard errors from the repeated trials (Tables 1 and 2, Fig 5) and in some cases relevant p-values for figures (Figure 3, 5). This data should be included as raw data, or ranges included on tables, or by appropriate statistical values.

Experimental design

The aim as stated by the authors was to determine the mode of action of PQQ4R in E. coli K12 wildtype, the dominant acrAB efflux pump mutant and acrAB overexpressed strain. The authors show that membrane disruption appears to be the primary mechanism of action but the precise targets of PQQ4R were not identified. This study provides incremental mechanistic details regarding the mechanism of PQQ4R mode of action as compared to previous studies. The merits of this study were the appropriate and in-depth comparative analysis of PQQ4R activity, efflux inhibition, ATP production, and cytotoxicity to two characterized efflux pump inhibitors (PABN and CPZ), making this study important to publish. The experimental results as presented appear sound and address key areas related to efflux pump inhibition by PQQ4R. The weaknesses of this article are the quality of writing, the lack of clear study rationale/focus and interpretation of results in some sections.
The introduction reads as is if it were cut and paste together and is missing vital information regarding what efflux pump inhibitors are, why the inhibitors were selected and why PQQ4R was selected for the study. A clear rationale and inclusion of the hypothesis buried within the results section would vastly improve the introduction.
The strength of the paper was the selection of experiments performed, and the results were described well enough to follow. The results often lacked necessary statistical analyses and required interpretation of the findings which are essential to understanding why the following experiments were performed. Why was the hypothesis of the paper presented in the middle of the results?
The discussion is overly brief, and the first two paragraphs include information that should have been provided in the introduction. Despite the toxic phenotype caused by PQQ4R, as stated in the results, the authors appear to recommend further development of this inhibitor. Would it also be more useful to elaborate upon how PQQ4R compares and contrasts with PABN and CPZ, as this was a major focus of the study?
The conclusions of the study seem to focus on PQQ4R’s potential as a therapeutic adjuvant. A more realistic focus related to PQQ4R’s actual merits or mode of action would be appropriate and match what was presented.

Detailed comments for each section:
Abstract
The abstract would benefit from a complete revision by focusing more clearly by summarizing the actual results obtained in this study. As currently written it is overly vague and does not highlight the main findings of the analysis nor the other EPIs characterized. For example, how PQQ4R compared to CPZ and PABN, MIC, efflux inhibition, membrane depolarization, ATP depletion, and cytotoxicity? The final sentence lines should either be omitted or entirely revised as it is highly misleading when compared to the actual findings stated by the authors.

Methods
Please clarify if OD 600 should have an “nm”. OD600 = Optical density 600 nm?

Please provide the genotype of AG100A in addition to AG100. Line 431 and Table 3 imply that this genotype includes the pleiotropic mutant tolC.

Efforts should be made to clarify and consistently use abbreviations within the text. All chemicals/substrates cited should be consistently abbreviated after their first in-text citation/usage eg. chlorpromazine (CPZ).

Results
Lines 255-257: Please describe why these efflux pump inhibitors were selected for comparison as this is an important part of the missing rationale for this study in the introduction.

Lines 262-263: This statement requires clarification. What is considered significant antimicrobial activity for efflux pump inhibitors? What was being compared in this statement? Why was AG100A the exception?

Line 264-265: “not only PQQ4R interacts with AcrAB but also might be a substrate of the AcrAB system” is a bold and potentially inaccurate conclusion without comparing MICs for PQQ4R to other efflux pump mutants. The authors should carefully reexamine their conclusions regarding PQQ4R MIC and avoid using the term “interact” until data can be provided to support this statement. Other efflux pumps (e.g. AcrD, AcrEF, and EmrAB) have been shown to influence efflux pump activity in the absence of AcrAB. It is difficult to confidently state the specificity of an interaction by AcrAB with PQQ4R/ PABN without systematically examining other tolC dependent/independent efflux pump mutants also present in E.coli K12 (mdfA, mdtABC, emrE, mdtK etc). The authors should avoid using the term “interact” until data can be provided to support this statement.

Lines 282-283: Caution should be taken by the authors when interpreting the 8-fold reduction caused by PQQ4R, strain already possess a significant growth phenotype and due to the transposition disruption, polar effects may also contribute to the PQQ4R induced phenotype.

Lines 285-286/ line 339-440/ Table 2: Why was tetracycline included in the MIC substrate testing for AG100tet in this study? The authors state that tetracycline resistance was the selectable marker used to overexpress acrAB by AG100tet? The phenotypes attributed to AG100tet tetracycline MICs are confounded and this issue should be stated in the text or the data omitted as obscures the phenotype caused by the efflux pump inhibitors

Lines 290-292: “The inability of PAβN to reduce MIC of ethidium bromide in wild-type strains has
already been described for E. coli and P. aeruginosa (Lomovskaya et al., 2001; Kern
292 et al., 2006).” Why was ethidium bromide selected if this was already known? What is the relevance of this data as compared to PQQ4R? An interpretation of the findings with respect to PQQ4R for the MIC assays would be appropriate here as this was the basis of this study. Ie. What did the MIC findings indicate about PQQ4R as compared to PABN and CPZ?

Line 304-305: “at non-antimicrobial concentrations” sub-inhibitory MIC concentrations would be a more appropriate term to use here.
Line 306/ Figure 1: The fluorescence intensity of equilibrium curves obtained for AG100A are significantly different from the wildtype or AG100tet. Based on the delay in time shown for AG100A and the growth phenotypes attributed to acrAB mutations, it would suggest that OD600nm =0.3 may not have been an appropriate physiology to select for this experiment. The cellular physiology of AG100A strain may have been too different for a proper comparison to the WT. A mid-log or late log OD value may have been more appropriate to use as lag phase physiology can cause a variety of delays in growth and significant alterations in gene expression which may influence the uptake and intercalation of EtBr. In whole cell experiments having all strains confidently entering the same growth phase may be more helpful in determining ethidium equilibrium concentrations. The authors should at minimum, provide growth curves comparing AG100, AG100A, and AG100tet in the media used to prepare these experiments. Alternatively, a technique like everted vesicle involving membranes isolated from these strains at the same protein concentrations would avoid some of the confounding physiological issues caused by using whole cell assays in this case. This may also explain why AG100A had much lower RFF values overall and a normalizing factor may be necessary between experiments to correct for these issues (refer to lines 319-322).
Figure 1. Please replace commas with decimal points cited for EtBr 0.25 ug/ml in panel B of Fig 1. Please include the uM concentration of EtBr used in these assays so molar comparisons of inhibitor to substrate can be made by the reader.
The authors should be cautious when drawing conclusions specifically related to findings related to acrAB only since the experiments primarily examined efflux pump inhibition by a broad efflux substrate ethidium. Ethidium is not only the substrate for multiple efflux pumps (emrAB, acrEF, mdfA, emrE) but its DNA intercalating/ mutagenic properties can cause additional mutations that can confound with efflux phenotypes.

Line 310-311/Table 3: How significant of a difference is the RFF value? What do low RFF values represent (accumulation with efflux)? What is the variance of the RFF values from three repeated assays as stated in lines 323-324? Is a RFF of 13.7 for PQQ4R significantly different as compared to PCZ and PABN? Why was PABN RFF so low? Also, please clarify what is meant by “indicating a strong ability to interfere with ethidium bromide efflux”?

How easily do PQQ4R and PABN penetrate these strains – is anything currently know about their diffusion/uptake? PABN was shown to permeabilize Gram-negative outer membranes as the authors cite in the discussion by Lamers et al. 2013. (http://dx.doi.org/10.1371/journal.pone.0060666). Was the concentration selected for PABN in the author’s experiments sufficient to avoid a concentration based outer membrane permeabilization effect for PABN? Or PQQ4R? How does this compare to PQQ4R and CPZ? Why were PABN and CPZ not included for experimental comparison to PQQ4R in Fig 3?

Figure 2: Please replace commas with decimals in the y-axis of all panels. Why was glucose not added to strains with efflux pump inhibitor? This would be useful to determine the extent of inhibition under physiologically relevant conditions.

Lines 319-326 are difficult to follow and would benefit from editing and clarification. Why was the RFF result expected for AG100tet? How overexpressed is acrAB in comparison to wildtype levels? How toxic is this overexpression, often artificial efflux pump overexpression causes growth delays and membrane depolarization/ destabilization.

Lines 326-327. The RFF values do not seem to support the statement made comparing PABN and CPZ values for AG100tet to the wildtype. Please clarify how RFF of 7.2 and 1.3 are similar to the wildtype?

Lines 331-332: What is a confirmatory efflux assay? How was this different than the assay performed above? Based on what was written the EtBr equilibrium assays were repeated with the addition of glucose. Please clarify what the conditions were in the statement “was subject to conditions” in line 332. Is this the addition of glucose only to the assay as alluded to in the following lines?

Lines 343-346 The hypothesis provided should be included in the last paragraph of the introduction which would greatly benefit from this a clear aim for this study.

Lines 369-370. Since PQQ4R has some membrane permeabilization effects at high concentration, would it not be prudent to confirm membrane integrity using an additional analysis such as K+ ion leakage assays at sub-inhibitory MICs tested (80 uM). These assays can be more sensitive to membrane permeabilizing drugs.

Line 404: Please provide MIC data for PQQ4R MIC assays as a supplementary figure/table. This information would make more sense to include with other MIC data in the first or second paragraph of the results.

Lines 420-421: What is meant by “an ON/OFF switch”? Please elaborate on this. Why is this worthy of note in cytotoxicity assays? This finding appears to contradict statements made in the abstract that downplay PQQ4R’s toxicity.

Discussion
Lines 424-435 The first paragraph of the discussion does not discuss the data and presented but provides some of the missing information necessary to include in the second last paragraph of the introduction. Please include this in the introduction.

Lines 440-441: the conclusions in this paragraph would be helpful to add in the results sections above.

Lines -456: Please elaborate on the statement: “This result is consistent with our hypothesis of an alternative mode of action in which at low concentrations PQQ4R inhibit E. coli efflux systems and near the MBC it interact directly with E. coli membranes.” How would such a mechanism work? Indirectly by local perturbations of lipid surrounding efflux pumps at low concentrations and membrane disruption at higher concentrations? How does this relate to summarized data in Fig 7?
In light of antimicrobial resistance mechanisms related to reduced drug permeability such as lipid modifications recently shown by colistin resistance (mcr-1), how useful would such an inhibitor be under these circumstances?

Lines 492-494. Dissipating bacterial PMF is a highly toxic approach for therapeutic approaches due to the extensive similarities of mitochondrial electron transport chain and its components. How would PQQ4R safely approach the concerns surrounding this? Specificity for select efflux pump(s)? Based on the current evidence presented, why should more energy and money be invested in developing these compounds?

Figure 7. What is the viability for PQQ4R at 160 uM (assumed 100% despite 85% membrane depolarization)?

Validity of the findings

The experimental findings appear to sound, appropriate for the aims of the study, and well conducted. The manuscript would benefit for some additional rationale for why some experiments were conducted and what the interpretation of the results was in the first 3 sections. The data is robust but lacks some statistical analysis which should be provided (refer to detailed comments in Experimental design).

---

## Round 0.2 · accepted · Accept

Based on the changes and responses you provided to each one of the reviewers comments, it was clear to me that your revised work was improved and respond to the quality we are looking for in PeerJ, therefore I invite you to look forward to our communications for its publication.

·

Basic reporting

The manuscript has been improved. The commentaries and critiques were addressed and added to the manuscript. References are appropriate. Figures are clear and the final message of the manuscript is supported by the experiments.

Experimental design

Experimental design is accurate.

Validity of the findings

Validity of the findings are well stated.

Additional comments

The authors answered the commentaries suggested. They performed new experiments (LDH determination) and also indicate the limitations (toxicity) of the molecule in the macrophage model. In addition they added other bacteria in the analysis.

Reviewer 3 ·

Basic reporting

The organization, background information, figures and tables in the revised manuscript are easier to follow and logically presented. Improvements to grammar, and sentence structure were noted and appreciated. The hypothesis is now clearly presented earlier in the manuscript making it easier to follow the results presented. The revised abstract now more accurately reflects the research described in the manuscript.

Experimental design

The research question, hypothesis, and background information in the revised version of the manuscript is improved. The authors have satisfactorily addressed the concerns and suggestions raised by each reviewer.The quality of the experiments remains high and now has sufficient detail and information for those interested in replicating the study.

Validity of the findings

The findings presented in the revised manuscript appear to be technically and statistically sound. Interpretation of findings and conclusions were easier to follow in the revised submission. All data presented are linked to the hypothesis presented.

Additional comments

The revised version of the manuscript has satisfactorily addressed the comments raised by each reviewing comment. The justifications made by the authors for not completing some suggested experiments/ revisions are logically presented and appear valid. For these reasons I see no reason not to recommend this manuscript for publication.